# Targeting MUC1-C Suppresses Chronic Activation of Cytosolic Nucleotide Receptors and STING in Triple-Negative Breast Cancer

**DOI:** 10.3390/cancers14112580

**Published:** 2022-05-24

**Authors:** Nami Yamashita, Atsushi Fushimi, Yoshihiro Morimoto, Atrayee Bhattacharya, Masayuki Hagiwara, Masaaki Yamamoto, Tsuyoshi Hata, Geoffrey I. Shapiro, Mark D. Long, Song Liu, Donald Kufe

**Affiliations:** 1Dana-Farber Cancer Institute, Harvard Medical School, 450 Brookline Avenue, D830, Boston, MA 02215, USA; nami_yamashita@dfci.harvard.edu (N.Y.); atsushi_fushimi@dfci.harvard.edu (A.F.); yoshihiro_morimoto@dfci.harvard.edu (Y.M.); atrayee_bhattacharya@dfci.harvard.edu (A.B.); masay.hagiwara@gmail.com (M.H.); masa0302g@gmail.com (M.Y.); tsuyoshihata1983@gmail.com (T.H.); geoffrey_shapiro@dfci.harvard.edu (G.I.S.); 2Department of Biostatistics & Bioinformatics, Roswell Park Comprehensive Cancer Center, Buffalo, NY 14263, USA; mark.long@roswellpark.org (M.D.L.); song.liu@roswellpark.org (S.L.)

**Keywords:** MUC1-C, TNBC, cGAS, STING, ISG15

## Abstract

**Simple Summary:**

Triple-negative breast cancers (TNBCs) are recalcitrant tumors with limited therapeutic options. Cytotoxic agents, including platinum-based drugs, are a standard of care for advanced TNBCs. Olaparib is also used for the treatment of germline *BRCA* mutant TNBC tumors in the adjuvant and recurrent disease settings. Notably, however, the effectiveness of these genotoxic agents is often limited by intrinsic and adaptive DNA damage resistance. We demonstrate in TNBC cells that the oncogenic MUC1-C protein chronically activates the type I interferon (IFN) pathway, drives the cGAS/STING axis and induces expression of the DNA damage resistance gene signature (IRDS). Targeting MUC1-C inhibits activation of this pathway in the response to carboplatin and olaparib and sensitizes TNBC cells to these agents. These findings indicate that MUC1-C is a target, which is druggable, for overcoming the obstacle of DNA damage resistance in the treatment of TNBCs.

**Abstract:**

The MUC1-C apical transmembrane protein is activated in the acute response of epithelial cells to inflammation. However, chronic MUC1-C activation promotes cancer progression, emphasizing the importance of MUC1-C as a target for treatment. We report here that MUC1-C is necessary for intrinsic expression of the RIG-I, MDA5 and cGAS cytosolic nucleotide pattern recognition receptors (PRRs) and the cGAS-stimulator of IFN genes (STING) in triple-negative breast cancer (TNBC) cells. Consistent with inducing the PRR/STING axis, MUC1-C drives chronic IFN-β production and activation of the type I interferon (IFN) pathway. MUC1-C thereby induces the IFN-related DNA damage resistance gene signature (IRDS), which includes ISG15, in linking chronic inflammation with DNA damage resistance. Targeting MUC1-C in TNBC cells treated with carboplatin or the PARP inhibitor olaparib further demonstrated that MUC1-C is necessary for expression of PRRs, STING and ISG15 and for intrinsic DNA damage resistance. Of translational relevance, MUC1 significantly associates with upregulation of STING and ISG15 in TNBC tumors and is a target for treatment with CAR T cells, antibody–drug conjugates (ADCs) and direct inhibitors that are under preclinical and clinical development.

## 1. Introduction

Mucin 1 (MUC1) is a heterodimeric protein that evolved in mammals to provide protection of epithelia from the external environment [1]. The MUC1 C-terminal subunit (MUC1-C) is a transmembrane effector of epithelial cell responses to inflammation and damage [1]. MUC1-C induces inflammatory, proliferative and remodeling signaling pathways that are associated with wound healing [1]. In this capacity, MUC1-C is activated by viral pathogens and plays a role in modulating the anti-viral inflammatory response [1]. However, in contrast to this acute protective response, prolonged activation of MUC1-C, as in settings of chronic inflammation with repetitive cycles of damage and repair, contributes to oncogenesis [1]. In this way, MUC1-C functions as an oncogenic protein and is aberrantly expressed in carcinomas [1]. MUC1-C induces hallmarks of the cancer cell, including the epithelial–mesenchymal transition (EMT), epigenetic reprogramming and cancer stem cell (CSC) state [2]. MUC1-C also promotes lineage plasticity in cancer progression by inducing pluripotency factors and by activating the nucleosome remodeling and histone deacetylase (NuRD), embryonic stem cell BAF (esBAF) and polybromo PBAF chromatin remodeling complexes [3,4,5,6,7]. In accordance with a role in driving lineage plasticity, MUC1-C contributes to cytotoxic and targeted drug resistance [1]. In addition, MUC1-C promotes immune evasion by chronic activation of the type II interferon-gamma pathway and the production of immunosuppressive effectors [8].

Pattern-recognition receptors (PRRs), which include the retinoic acid-inducible gene-1 (RIG-I/DDX58) and melanoma differentiation-associated gene 5 (MDA5/IFIH1), are activated by the presence of cytosolic dsRNA [9,10]. Accumulation of DNA in the cytosol is recognized by activation of the cyclic GMP-AMP (cGAMP) synthase (cGAS)-stimulator of IFN genes (STING) pathway [11,12]. The RIG-I and MDA5 cytosolic RNA sensors can also function as nonredundant cytosolic DNA sensors [13,14]. Stimulation by “non-self” cytosolic nucleotide, pathogen-associated molecular patterns (PAMPs), induces production of the type I IFNs (IFN-α, IFN-β) and activation of innate immune responses [10,15,16]. In cancer cells, the type I IFN pathway is also activated by DNA damage-associated molecular patterns (DAMPs) that are generated in part by DNA leakage from the nucleus in association with genomic instability and the upregulated expression of double-stranded RNAs that are encoded by endogenous retrovirus-like DNA sequences [10,15,16]. The type I IFNs activate the downstream canonical STAT2 and interferon regulatory factor 9 (IRF9) pathway to induce interferon-stimulated genes (ISGs) [10,15,16]. Chronic exposure to low levels of type I IFNs and activation of the type I IFN pathway by DAMPs in cancer cells has been linked to oncogene-induced replicative stress [17,18] and induction of the IFN-related DNA damage resistance signature (IRDS) [15,19,20]. IRDS genes encode STAT1, as well as ISG15, which is overexpressed in diverse cancers and links chronic inflammation with DNA damage resistance [21,22,23]. Notably, dysregulation of ISG15 in cancer cells promotes the accumulation of cytosolic DNA, activation of cGAS-STING and chronic induction of type I IFN signaling in a potential auto-inductive loop [22,24].

MUC1-C evolved as an acute effector of the inflammatory response in normal epithelia [1]; however, there is no known association between MUC1-C and DAMP-induced chronic activation of type I IFN pathway in cancer cells. In this respect, the present work focuses on TNBC cells and specifically the regulation of type I IFN RIG-I, MDA5, cGAS and STING effectors. Our analysis of datasets derived from TNBC tumors demonstrate that MUC1 is significantly associated with expression of STAT1, STAT2, IRF9, STING and ISG15. In addition, studies in TNBC cell lines show that MUC1-C is necessary for induction of (i) the RIG-I, MDA5 and cGAS PRRs, (ii) STING and (iii) IFN-β-mediated chronic activation of the IRDS pathway. We also report that MUC1-C is necessary for the expression of ISG15 and that TNBC cells are addicted to the MUC1-C→ISG15 signaling pathway. Importantly, targeting MUC1-C in TNBC cells treated with carboplatin or the PARP inhibitor olaparib blocks activation of the PRR/STING axis and intrinsic DNA damage resistance.

## 2. Results

MUC1 expression in TNBC tumors associates with type I IFN and inflammatory pathways.

Using the TCGA-BRCA transcriptome dataset, we found by GSEA of TNBC tumors that MUC1 significantly associates with activation of the HALLMARK INTERFERON ALPHA RESPONSE pathway (Figure 1A). Similar results were obtained from GSEA of TNBC tumors in the METABRIC dataset (Figure 1B). Type I IFNs activate the IFN-stimulated gene factor 3 (ISGF3) complex, which induces genes containing IFN-stimulated response elements (ISREs) [25]. ISGF3 function is dependent on the interaction between STAT2 and IRF9 [25,26]. Consistent with activation of the type I IFN pathway, we found that MUC1 significantly associates with (i) STAT2 and IRF9 expression in TNBC tumors (Figure 1C) and (ii) IRF9 in a scRNA-seq dataset from TNBC tumor cells (Figure 1D). These findings were extended by demonstrating that MUC1-high TNBCs are significantly enriched for type I IFN response genes encoding, among others, the (i) IFN-induced protein with tetratricopeptide repeats (IFIT2/3) proteins [27], (ii) oligoadenylate synthase (OAS) 2 that senses viral dsRNAs and synthesizes 2′-5′-linked oligoadenylates [28], and (iii) OAS-like protein (OASL), which is a potent regulator of anti-viral responses functioning in a positive circuit with RIG-I [29] (Figure 1E). Further analysis of the TCGA-BRCA TNBC (Figure 1F) and METABRIC TNBC (Figure 1G) datasets demonstrated that MUC1 activates the HALLMARK INFLAMMATORY RESPONSE, indicating that MUC1 associates with chronic inflammation in TNBC cells.

### 2.1. MUC1-C Induces RIG-I and MDA5 Cytosolic RNA PRRs in TNBC Cells

As found from analysis of the TCGA-BRCA and METABRIC TNBC datasets, GSEA of RNA-seq signatures from BT-549 TNBC cells demonstrated that MUC1-C significantly associates with the HALLMARK INTERFERON ALPHA RESPONSE (Figure 2A; Appendix A), HALLMARK INFLAMMATORY RESPONSE (Figure 2B; Appendix A), REACTOME INTERFERON ALPHA BETA SIGNALING (Appendix A) and GO RESPONSE TO TYPE I INTERFERON (Appendix A) pathways. Type I IFN signaling is activated in part by the RIG-I and MDA5 cytosolic RNA sensing PRRs [9,30]. Here, we found that silencing MUC1-C significantly associates with downregulation of the KEGG RIG-I LIKE RECEPTOR SIGNALING PATHWAY (Figure 2C; Appendix A) and the REACTOME DDX58 IFIH1 MEDIATED INDUCTION OF INTERFERON ALPHA BETA gene signatures (Figure 2D; Appendix A). We also found that inducible and stable MUC1-C silencing in BT-549 cells suppresses expression of RIG-I and MDA5 (Figure 2E,F, left and right). As confirmation of these results, rescue of MUC1-C silencing with the MUC1-C cytoplasmic domain (MUC1-CD) restored RIG-I and MDA5 expression (Appendix A). Similar findings were obtained in MDA-MB-436 *BRCA1* mutant TNBC cells; that is, MUC1-C-dependent induction of RIG-I and MDA5 (Appendix A), indicating that their activation is not restricted to a single TNBC cell type or *BRCA1* status. To further define MUC1-C involvement in triggering the type I IFN pathway by RNA sensing, we transfected BT-549 cells with poly I:C to mimic pathogen-derived dsRNA [9] and found that MUC1-C is necessary for poly I:C-induced expression of RIG-I and MDA5 (Figure 2G). Type I IFN production is subject to positive feedback regulation in promoting chronic inflammation [18]. Along these lines, we stimulated cells with IFN-β and found that induction of RIG-I and MDA5 is MUC1-C-dependent (Figure 2H), indicating that MUC1-C plays a role in chronic activation of the type I IFN pathway.

### 2.2. MUC1-C Activates the cGAS/STING Cytosolic DNA Sensing Pathway in TNBC Cells

As identified for cytosolic RNA sensing, silencing MUC1-C in BT-549 cells was significantly associated with downregulation of the CYTOSOLIC DNA SENSING PATHWAY (Figure 3A; Appendix A). cGAS is a central regulator of cytosolic DNA sensing, which induces type I IFN production through the stimulation of STING [12,31]. We found that inducible and stable silencing of MUC1-C in BT-549 cells suppresses (i) cGAS and STING mRNA levels (Appendix A) and (ii) cGAS and STING proteins (Figure 3B, left and right), which was rescued by restoring MUC1-CD expression (Appendix A). In addition, MUC1-C was necessary for expression of cGAS and STING transcripts and proteins in *BRCA1* mutant MDA-MB-436 (Figure 3C; Appendix A) and SUM149 (Figure 3D; Appendix A) *BRCA1* mutant TNBC cells indicating that this pathway is also independent of *BRCA1* status. Analysis of TNBC tumors in the TCGA-BRCA dataset further demonstrated that MUC1 significantly associates with STING, but not cGAS, expression (Figure 3E). Along these lines, we found that MUC1-C is necessary for activation of STING in the response of BT-549 cells to poly dA:dT transfection (Figure 3F, left) and IFN-β stimulation (Figure 3F, right). Moreover, MUC1-C was necessary for expression of IFN-β transcripts (Figure 3G; Appendix A) and production (Figure 3H) in support of involvement in activating a potential STING/IFN-β auto-inductive pathway.

### 2.3. MUC1-C Drives U-ISGF3 Target DNA Damage Resistance Genes in TNBC Cells and Tumors

Chronic IFN-β stimulation of the type I IFN pathway is mediated by the unphosphorylated ISGF3 (U-ISGF3) complex that includes STAT2 and IRF9 [32]. In concert with the results from TNBC tumors, we found that inducible and stable MUC1-C silencing decreases expression of U-STAT2 and IRF9, as well as U-STAT1, which also contributes to the U-ISGF3 complex (Figure 4A,B, left and right). The U-ISGF3 complex activates genes that mediate resistance to viruses and DNA damage [15,16,19,32,33]. Silencing MUC1-C resulted in the marked downregulation of these U-ISGF3 target genes (Figure 4C). Among those associated with the IRDS [34], we confirmed that MUC1-C is necessary for expression of (i) IRF7, which amplifies constitutive IRF3 activation and the synthesis of type I IFNs [35,36], (ii) oligoadenylate synthase (OAS) 1 and OAS3 that sense viral dsRNAs and synthesize 2′-5′-linked oligoadenylates [28,37], (iii) IFIT1 and MX1 proteins that promote innate immunity and DNA damage resistance [27,38] and (iv) interferon-induced transmembrane 1 (IFITM1), which is overexpressed in carcinomas and contributes to therapeutic resistance [39] (Figure 4D). Analysis of MDA-MB-436 and SUM149 cells further demonstrated that MUC1-C is necessary for IRF7, IFIT1, MX1 and IFITM1 expression (Appendix A). Underscoring these results, analysis of bulk RNA-seq TNBC tumor datasets demonstrated that MUC1 expression significantly associates with expression of IRDS genes (Figure 4E,F). Analysis of scRNA-seq TNBC data further supported the significant association between MUC1 and expression of IRDS genes encoding IRF7, BST2, IFI35 and IFITM1 in individual TNBC tumor cells (Figure 4G).

### 2.4. MUC1-C Interacts with STAT1 and IRF1 in Chronic IFN-β Stimulation of the Type I IFN Pathway

STAT1 and IRF1 are important effectors of chronic IFN-β stimulation, which have the capacity to complement each other in the activation of ISREs [32,35,40]. We found that MUC1-C is necessary for both constitutive and IFN-β-mediated induction of STAT1 and IRF1 expression (Figure 5A, left and right). Moreover, we found that MUC1-C forms nuclear complexes with STAT1 and IRF1 in the response to IFN-β stimulation (Figure 5B). In support of the functional significance of MUC1-C/STAT1/IRF1 complexes and as shown for MUC1-C, silencing STAT1 and IRF1 (Appendix A) suppressed (i) RIG-I, MDA5 and STING (Figure 5C), and (ii) IRDS (Appendix A) expression. In further dissecting this potential mechanism, we performed ChIP studies on the *RIG-I* promoter and found that occupancy of the ISRE by MUC1-C, STAT1 and IRF1 is suppressed by MUC1-C silencing (Figure 5D). Like *RIG-I*, the *MDA5* promoter region includes an ISRE that is occupied by MUC1-C, STAT1 and IRF1 in a MUC1-C-dependent manner (Figure 5E). Similar results were obtained with studies of a proximal region downstream to the *STING* TSS (Figure 5F), supporting a model in which MUC1-C promotes STAT1 and IRF1 occupancy on ISREs in driving *RIG-I*, *MDA5* and *STING* gene expression.

### 2.5. MUC1-C Is Necessary for Induction of ISG15 Expression

The IRDS also includes the *ISG15* gene [15,19], which encodes a ubiquitin-like protein that functions as a major effector of innate immunity and DNA damage resistance [24,41]. We found that MUC1-C is necessary for induction of ISG15 in the response to stimulation with IFN-β (Figure 6A, left and right). Moreover, STAT1 and IRF1 were necessary for ISG15 expression (Figure 6B). The *ISG15* promoter contains two ISREs (Figure 6C) [35,42]. ChIP analysis of the *ISG15* promoter region showed that occupancy of MUC1-C, STAT1 and IRF1 is MUC1-C dependent (Figure 6C), consistent with involvement of MUC1-C in inducing ISG15 expression. ISG15 plays a key role in the DNA damage response (DDR) by promoting translesion DNA synthesis, a DNA damage tolerance mechanism which allows cells to bypass DNA lesions that block replication [21,23,24]. As a result, cells tolerate the repair of these lesions at a later stage and avoid the collapse of replication forks [21,23,24]. Consistent with these functions, downregulation of ISG15 is associated with pronounced reductions in replicative track length [24]. By extension, we found that (i) inducible and stable silencing of MUC1-C with downregulation of ISG15 (Figure 6D, left and right), which is rescued by MUC1-CD (Appendix A), (ii) targeting MUC1-C with the GO-203 inhibitor (Appendix A) [1] and (iii) silencing ISG15 (Figure 6E) result in induction of γH2AX, PARP1 cleavage and apoptotic cell death (Appendix A–G), in support of addiction to the MUC1-C→ISG15 pathway. MUC1 and ISG15 are overexpressed in diverse cancers [1,21,23]. In concert with a MUC1-C→ISG15 pathway, analysis of the TCGA-BRCA and METABRIC datasets demonstrated that MUC1 is significantly associated with ISG15 expression in TNBC tumors (Figure 6F). Moreover, and of potential translational relevance, targeting MUC1-C represses ISG15 in association with abrogation of DNA damage tolerance and induction of DNA damage-induced apoptosis.

### 2.6. MUC1-C Is Necessary for Induction of PRRs, STING and ISG15 in the Response to Treatment with Carboplatin and Olaparib

Conventional chemotherapy remains the standard of care for the treatment of TNBCs without germline *BRCA* mutations; however, the effectiveness of platinum-based agents has been limited by the development of DNA damage resistance [43,44]. MUC1-C is necessary for the repair of DNA damage in TNBC cells treated with cisplatin, consistent with a role in promoting DNA damage resistance [45]. Given that MUC1-C is necessary for constitutive cGAS/STING expression, we therefore asked whether MUC1-C is also involved in the activation of PRRs and STING in the response of BT-549 cells to carboplatin treatment. As expected, we found that carboplatin induces the upregulation of RIG-I, MDA5 and cGAS, but not STING in this TP53 mutant background (Figure 7A). In addition, we found that MUC1-C is necessary for the expression of RIG-I, MDA5, cGAS, STING and ISG15, as well as γH2AX, in the response to carboplatin-induced DNA damage (Figure 7A). MUC1-C localizes to the nucleus in the response of TNBC cells to genotoxic agents [45]. Here, γH2AX staining was performed as a direct measurement of DNA stability. Immunocytochemistry (ICC) images demonstrated that γH2AX staining is increased with (i) MUC1-C silencing, (ii) carboplatin treatment and (iii) their combination (Figure 7B). Higher magnification showed that γH2AX staining is largely a small nuclear foci and not an apoptotic γH2AX nuclear ring (Figure 7C), indicating that induction of γH2AX is associated with activation of the DDR and not apoptosis [46]. Treatment of *BRCA* mutant TNBCs with PARP inhibitors (PARPi) has also been limited by primary and acquired resistance [43,44,47]. Targeting MUC1-C synergistically sensitizes both *BRCA* wild type and mutant TNBC cells to the PARPi olaparib [45]. In MDA-MB-436 cells, olaparib promotes DNA damage, increases accumulation of DNA in the cytosol and induces STING expression [48]. By extension and consistent with the carboplatin studies, we found that MUC1-C is necessary for expression of RIG-I, MDA5, cGAS, STING and ISG15 in olaparib-treated MDA-MB-436 cells (Figure 7D). Moreover, targeting MUC1-C attenuated DNA damage resistance as evidenced by increases in γH2AX expression (Figure 7D) and detection of γH2AX nuclear foci (Figure 7E,F). Based on these results, we asked whether targeting MUC1-C is sufficient to downregulate the PRR/STING axis such that it would not be able to exert a pro-survival function when the cytosolic nucleotide sensing pathway is induced by genotoxic insults. Along these lines, we found that cell viability following treatment with carboplatin (Figure 7G) or olaparib (Figure 7H) is significantly reduced over time in MUC1-C silenced cells as compared to control cells, further indicating that MUC1-C confers resistance to DNA damage. These findings collectively indicate that MUC1-C is essential for activation of the PRR/STING axis in the response of TNBC cells to genotoxic agents and that targeting MUC1-C attenuates DNA damage resistance.

## 3. Discussion

The present work demonstrates that MUC1-C, which evolved in mammals to protect epithelia in part from viral pathogens [1], functions in chronic activation of the type I IFN pathway in TNBC cells. We found that MUC1-C constitutively induces the RIG-I and MDA5 cytosolic RNA receptors, which are activated by the abnormal accumulation of endogenous RNAs in what has been defined as “RNA stress” [30]. RIG-I and MDA5 can also function in a nonredundant way as cytosolic DNA sensors [13]. Our results further demonstrate that MUC1-C is necessary for constitutive expression of cGAS, which is activated by the cytosolic accumulation of DAMPs that are the products of genomic instability in cancer cells [10,11,12] (Figure 7C). Mechanistically, we show that (i) MUC1-C associates with STAT1 and IRF1, and (ii) MUC1-C/STAT1/IRF1 signaling is necessary for induction of RIG-I, MDA5 and cGAS/STING expression. MUC1-C is activated in the response of epithelial cells to RNA and DNA viral infections, which is a reversible process with resolution of inflammation and reestablishment of homeostasis [1]. Our findings that MUC1-C constitutively induces PRRs in TNBC cells suggest that MUC1-C has exploited chronic activation of the type I IFN pathway to promote TNBC progression. In this respect, acute exposures to IFNs suppress survival of cancer cells, whereas chronic low levels of IFN production induce U-ISGF3-mediated ISGs and the IRDS to provide protection against replicative stress [16]. Along these lines, we found that MUC1-C is necessary for intrinsic IFN-β production in TNBCs. In addition, MUC1-C promotes genomic instability and the type I IFN pathway is chronically activated by oncogene-induced replicative stress [17,18]. Together, these findings suggest that MUC1-C has exploited activation of PRRs and the type I IFN pathway to promote induction of the IRDS and thereby protect against replication stress. This involvement of MUC1-C in the replication stress response also has the capacity to promote the accumulation of mutations in downstream effectors that contribute to cancer progression. Indeed, prolonged MUC1-C activation in settings of chronic inflammation promotes oncogenesis [1].

Activation of PRRs induces the production of the type I IFNs in promoting anti-viral responses [10]. Consistent with chronic activation of this pathway in TNBC cells, MUC1-C was necessary for constitutive IFN-β production and downstream induction of the STAT2 and IRF9 effectors of the U-ISGF3 complex that activates ISGs. As a result, MUC1-C was also necessary for expression of (i) OAS1, which attenuates PAR synthesis during DNA repair and promotes the ability of cancer cells to survive replicative stress [37], and (ii) IFIT1-3, MX1 and BST2, among others, which confer DNA damage resistance [34]. In support of these findings, MUC1 was significantly associated with IFIT2/3, OAS2/3, OASL, IFI35 and IFITM1 expression in TNBC tumors. Of relevance for shedding light on why MUC1-C is necessary for induction of these ISGs, MUC1-C promotes DNA damage resistance by inducing the repair of DSBs in the response to genotoxic anti-cancer agents [1,45,49]. In line with this capacity, the IRDS is associated with resistance to chemotherapy and/or radiation across diverse cancer cell types and confers poor patient outcomes [19]. The present work extends those observations by demonstrating that MUC1-C is also necessary for the induction of ISG15 expression. ISG15 was initially identified as a component of anti-viral immunity; however, subsequent work demonstrated a role for this ubiquitin-like protein in modulation of p53 signaling and replication fork progression [23]. In this way, ISG15 regulates replication fork speed, which is necessary for ensuring genomic stability [23,24]. ISG15 is upregulated in diverse cancers by unclear mechanisms and is of potential importance for their responsiveness to genotoxic agents [23]. Our results demonstrate that MUC1-C in a complex with STAT1 and IRF1 constitutively activates *ISG15* transcription, in support of MUC1-C involvement in integrating chronic activation of the type I IFN pathway with induction of ISG15 expression in cancer cells. The MUC1-C→ISG15 pathway therefore has the capacity for conferring DNA damage resistance.

The significance of the MUC1-C→ISG15 pathway in TNBC cells is further supported by the demonstration that (i) MUC1 is significantly associated with upregulation of ISG15 in TNBC tumors and (ii) downregulation of MUC1-C suppresses ISG15 expression. In concert with the involvement of ISG15 in regulating replication fork speed, decreases in ISG15 reduce replicated track length [24]. We found that silencing ISG15 in TNBC cells is associated with induction of DNA damage and apoptosis, indicating that reductions in track lengths may be potentially lethal lesions. As a result, targeting MUC1-C and thereby downregulating ISG15 induces DNA damage and the apoptotic response. These findings, while opening the way for more mechanistic studies, suggest that TNBC and perhaps other types of cancer cells may be addicted to MUC1-C→ISG15 signaling as a facilitator of DNA damage tolerance [21,23,24]. MUC1-C-induced ISG15 expression thus expands the inextricable links among intrinsic activation of the type I IFN pathway, DNA damage resistance and chronic inflammation in cancer cells. In this context, chronic DAMP-induced PRR/STING activation leads to persistent low-dose IFN-β autocrine and paracrine stimulation of cancer cells, which induces UISGF3-mediated transcription of the non-canonical pathway that overlaps with the IRDS (STAT1, ISG15) and includes RIG-I and MDA5 [32]. Sustained transcription of STAT1, RIG-I and MDA5 thus establishes a MUC1-C→PRR/STING→STAT1/RIG-I/MDA5→PRR/STING auto-inductive circuit. Along these lines, we found that MUC1-C induces intrinsic STAT1- and IRF1-dependent expression of STING. Additionally, as shown for MUC1-C [1], chronic activation of STING has been linked to persistent inflammation and immunosuppression [50,51]. MUC1-C and STING also share characteristics, such as promoting DNA damage resistance [52], that contribute to chronic inflammation and other hallmarks of the cancer cell [1,12]. As an alternative pathway, RIG-I can mediate inflammatory signaling by cGAS-independent mechanisms, which include the induction of STING expression and direct interactions with STING [53]. Our findings hold other potentially significant implications in that treatment of TNBC cells with carboplatin and olaparib is dependent on MUC1-C for activation of RIG-I, MDA5, cGAS, STING and ISG15, indicating that targeting MUC1-C can block this compensatory response to increases in replicative stress.

## 4. Materials and Methods

### 4.1. Cell Culture

Human BT-549 *BRCA1* wild type TNBC (American Type Culture Collection (ATCC), Manassas, VA, USA) cells were cultured in RPMI1640 medium (Thermo Fisher Scientific, Waltham, MA, USA) containing 10% fetal bovine serum (FBS; GEMINI Bio-Products, West Sacramento, CA, USA), 100 μg/mL streptomycin, 100 U/mL penicillin and 10 μg/mL insulin. MDA-MB-436 *BRCA1* mutant TNBC (ATCC) cells were cultured in Leibovitz’s L-15 medium (Thermo Fisher Scientific) containing 10% FBS with 10 μg/mL insulin and 16 μg/mL glutathione. SUM149 *BRCA1* mutant TNBC (ATCC) cells were grown in Ham’s F-12 medium (Corning, Manassas, VA, USA) supplemented with 10 mM HEPES, 5% FBS, 100 μg/mL streptomycin, 100 U/mL penicillin, 5 μg/mL insulin and 1 μg/mL hydrocortisone. Cells were transfected with 1 μg/mL of poly (I:C) LMW or poly (dA:dT) (InvivoGen, San Diego, CA, USA) in the presence of Lipofectamine 3000 Reagent (Thermo Fisher Scientific). Cells were treated with (i) the MUC1-C inhibitor GO-203 [4], (ii) 0.1 ng/mL human recombinant IFN-β (STEMCELL Technologies, Vancouver, BC, Canada), (iii) 20 μM carboplatin (C2538; Millipore Sigma, St. Louis, MO, USA) and (iv) 5 μM olaparib (AZD2281; Selleck Chemicals, Houston, TX, USA). Cell authentication was performed by short tandem repeat analysis. Cells were monitored for mycoplasma contamination using the MycoAlert Mycoplasma Detection Kit (Lonza, Rockland, MA, USA).

### 4.2. Gene Silencing and Rescue

MUC1shRNA (MISSION shRNA TRCN0000122938) and a control scrambled shRNA (CshRNA) (Millipore Sigma) were inserted into pLKO-tet-puro (Plasmid #21915; Addgene, Cambridge, MA, USA). The MUC1-C cDNA was inserted into the pinducer20 vector (Plasmid #44012; Addgene). The CshRNA, MUC1shRNA, MUC1shRNA#2 (MISSION shRNA TRCN0000430218), STAT1shRNA (MISSION shRNA TRCN0000004266), IRF1shRNA (MISSION shRNA TRCN0000014672), ISG15shRNA (MISSION shRNA TRCN0000007420) and ISG15shRNA#2 (MISSION shRNA TRCN0000237825) were produced in HEK293T cells as described [4]. Flag-tagged MUC1-CD [49] was inserted into pInducer20 (Plasmid #44012, Addgene) [54]. Cells transduced with the vectors were selected for growth in 1–2 μg/mL puromycin or 100 μg/mL geneticin. For inducible gene silencing, cells were treated with 0.1% DMSO as the vehicle control or 500 ng/mL doxycycline (DOX; Millipore Sigma).

### 4.3. Real-Time Quantitative Reverse-Transcription PCR (qRT-PCR)

Total RNA was isolated using Trizol reagent (Invitrogen, Carlsbad, CA, USA) as described [8]. cDNAs were synthesized using the High Capacity cDNA Reverse Transcription Kit (Applied Biosystems, Grand Island, NY, USA) as described [8]. Samples were amplified using the Power SYBR Green PCR Master Mix (Applied Biosystems) and the CFX96 Touch Real-Time PCR Detection System (Bio-Rad Laboratories, Hercules, CA, USA) as described [8]. Primers used for qRT-PCR analysis are listed in the Appendix A.

### 4.4. Immunoblot Analysis

Whole cell lysates were prepared in RIPA buffer containing protease inhibitor cocktail (Thermo Fisher Scientific, Waltham, MA, USA). Immunoblotting was performed with anti-MUC1-C (#16564, 1:1000 dilution; Cell Signaling Technology (CST), Danvers, MA, USA), anti-RIG-I (#3743, 1:1000 dilution; CST), anti-MDA5 (#5321, 1:1000 dilution; CST), anti-cGAS (#15102, 1:1000 dilution; CST), anti-STING (#13647, 1:1000 dilution; CST), anti-STAT1 (#9172, 1:1000 dilution; CST), anti-STAT2 (#72604, 1:1000 dilution; CST), anti-IRF1 (#8478, 1:1000 dilution; CST), anti-IRF9 (#76684, 1:1000 dilution; CST), anti-ISG15 (sc-166755, 1:250 dilution; Santa Cruz, Santa Cruz, CA, USA), anti-PARP1 (#9532, 1:1000 dilution; CST), anti-γH2AX (#9718, 1:1000 dilution; CST) and anti-β-actin (A5441; 1:50000 dilution; Sigma, St. Louis, MO, USA).

### 4.5. Coimmunoprecipitation of Nuclear Proteins

Nuclear lysates were isolated as described [45]. DNA was digested by incubation in 20 U/mL DNase for 30 min at 37 °C as described [45]. Nuclear proteins were incubated with anti-MUC1-C (#MA5-11202, Thermo Fisher Scientific) at 4 °C overnight and then precipitated with Dynabeads Protein G (10003D; Thermo Fisher Scientific) for 2 h at 4 °C. Beads were washed and then resuspended in a sample loading buffer as described [45].

### 4.6. Chromatin Immunoprecipitation (ChIP)

ChIP was performed on cells crosslinked with 1% formaldehyde for 5 min at 37 °C, quenched with 2 M glycine and washed with PBS, and then sonicated in a Covaris E220 sonicator to generate 300–600 bp DNA fragments as described [8]. Immunoprecipitation was performed using a control IgG (Santa-Cruz Biotechnology) and antibodies against MUC1-C (#MA5-11202, Thermo Fisher Scientific), STAT1 (#9172, CST), IRF1 (#8478, CST)**.** Precipitated DNAs were detected by PCR using primers listed in Appendix A. Quantitation was performed on immunoprecipitated DNA using SYBR-green and the CFX96 Touch Real-Time PCR Detection System (Bio-Rad, Hercules, CA, USA). Data are reported as fold enrichment relative to IgG [4].

### 4.7. ELISA for Measurement of IFN-β Levels

IFN-β concentrations were quantified with Human IFN-beta Quantikine QuicKit ELISA (R&D systems, Minneapolis, MN, USA). ELISAs were performed according to the manufacturer’s instructions on supernatants collected from cells after 72 h in culture.

### 4.8. Cell Proliferation Assays

Cell viability was assessed using the Alamarblue assay (Thermo Scientific, Rockford, IL, USA). Absorbance (570 nm) or fluorescence intensity (560 nm excitation/590 nm emission) was measured in sextuplicate wells.

### 4.9. Apoptosis Assays

Cells were harvested and stained with the Dead Cell Apoptosis Kit with Annexin V Alexa Fluor 488 and propidium iodide (V13241, Thermo Scientific, Rockford, IL, USA). The cell apoptosis ratio was measured according to the manufacturer’s instructions by flow cytometry.

### 4.10. ICC Analysis of γH2AX Expression

BT-549 and MDA-MB-436 cells were fixed in 4% paraformaldehyde (Sigma) at room temperature for 10 min. The samples were incubated with 0.1% Triton X-100 (Sigma) at room temperature for 10 min, blocked with 3% Normal Goat Serum (Gibco), incubated with anti-γH2AX (#9718, 1:400 dilution; Cell Signaling Technology) at 4 °C overnight and then incubated with goat anti-rabbit IgG H and L labeled with Alexa Fluor 488 (Abcam) at room temperature for 1 h. Invitrogen™ ProLong™ Gold Antifade Mountant with DAPI (Invitrogen) was used for staining of nuclei. The cells were analyzed using a Leica THUNDER Imager 3D Cell Culture microscope.

### 4.11. Statistical Analysis

Experiments were repeated at least three times and data are expressed as the mean ± SD as described [8]. The unpaired Student’s *t*-test or Wilcoxon rank sum test were used to examine differences between means of two groups. A *p*-value of <0.05 was considered a statistically significant difference.

### 4.12. Analysis of Publicly Available TNBC Cohort and scRNA-Seq Datasets

TCGA-BRCA expression and clinical annotations were obtained from the Genomic Data Commons (GDC) data portal, processed via TCGAbiolinks package in R using TCGA Workflow guided practices and analyzed as described [8]. Data from the publicly available GSE118390 scRNA-seq dataset of TNBC samples [55] were obtained directly from GEO via *GEOquery* and analyzed as described [8].

### 4.13. Data and Software Availability

The accession number for the RNA-seq data reported in this paper is GEO ACCESSION GSE164141.

## 5. Conclusions

In summary, our findings that MUC1-C drives chronic activation of PRRs, STING, the type I IFN pathway and the IRDS in TNBC cells are of potential importance to the regulation of the replication stress response, which has been linked to chronic inflammation, DNA damage resistance and cancer progression. Our results further demonstrate that MUC1-C is a potential target for abrogating this pathway of DNA damage tolerance in cancer cells. MUC1-C promotes chronic activation of the type II interferon-gamma pathway and the production of immunosuppressive effectors [8]. Along with the present work, these findings collectively indicate that MUC1-C-induced chronic activation of both the type I and II IFN pathways, integrates immunosuppression and DNA damage resistance, which are widely recognized as being associated with the CSC state. Of further importance, agents targeting MUC1-C, which include CAR T cells, ADCs, GO-203 and anti-sense oligonucleotides (ASOs), are in preclinical and clinical development for suppressing DNA damage resistance in the treatment of patients with TNBC [1]. Based on the present findings, these anti-MUC1-C agents could have the potential for increasing the sensitivity of advanced TNBCs to genotoxic drugs, as well as *BRCA* mutant TNBC tumors to olaparib in the adjuvant and recurrent disease settings.

## Figures and Tables

**Figure 1 cancers-14-02580-f001:**
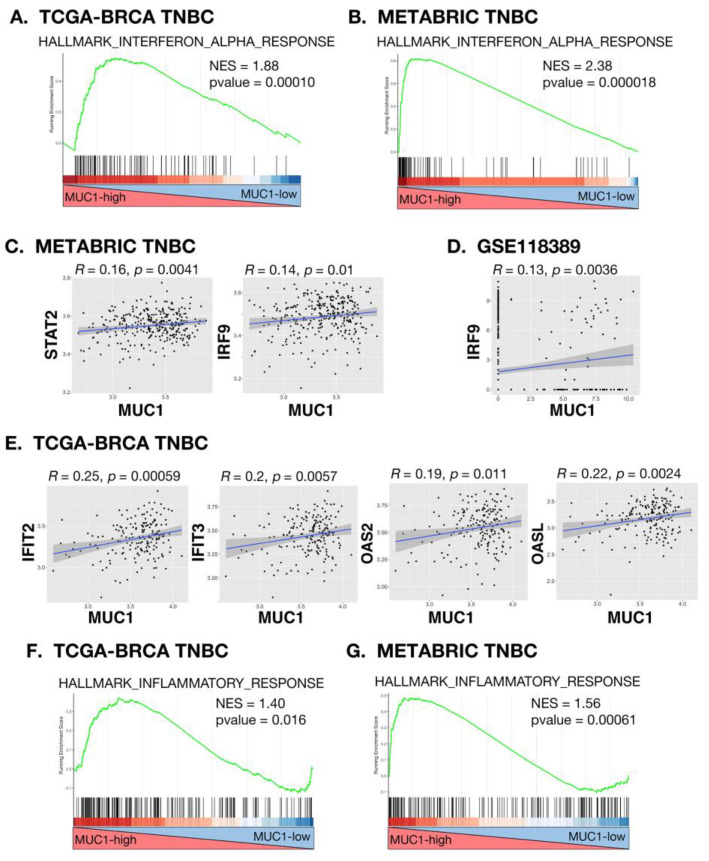
MUC1-high TNBC tumors associate with type I IFN and inflammatory signaling pathways. (**A**,**B**) Enrichment plots for the HALLMARK INTERFERON ALPHA RESPONSE pathway, comparing MUC1-high to MUC1-low TNBC tumors in the TCGA-BRCA (**A**) and METABRIC (**B**) cohorts. (**C**) Scatter plots showing correlations of MUC1 with STAT2 and IRF9 in TNBCs from the METABRIC cohort. (**D**) Scatter plot of the correlation between MUC1 and IRF9 in single TNBC cells as analyzed from the GSE118389 scRNA-seq dataset. (**E**) Scatter plots showing correlations of MUC1 with IFIT2, IFIT3, OAS2 and OASL in TNBCs from the TCGA-BRCA and METABRIC TNBC cohorts. (**F**,**G**) Enrichment plots for the HALLMARK INFLAMMATORY RESPONSE pathway, comparing MUC1-high to MUC1-low TNBC tumors in the TCGA-BRCA (**F**) and METABRIC (**G**) cohorts.

**Figure 2 cancers-14-02580-f002:**
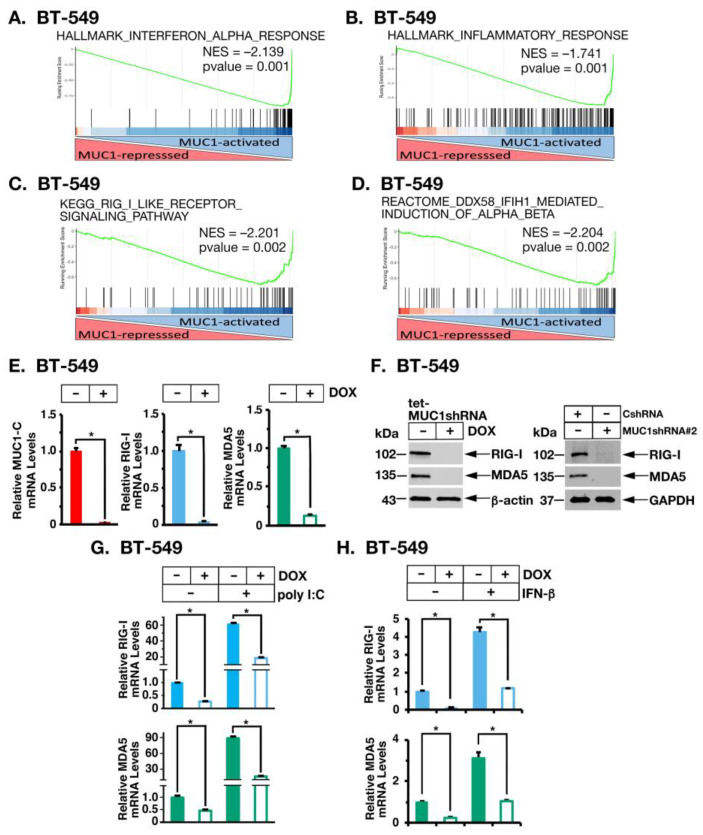
MUC1-C is necessary for induction of RNA PRRs. (**A**–**D**) RNA-seq was performed in triplicate on BT-549 cells expressing a tet-MUC1shRNA, which were treated with vehicle or DOX for 7 days. Candidate pathway enrichment plots for the HALLMARK INTERFERON ALPHA RESPONSE (**A**), HALLMARK INFLAMMATORY RESPONSE (**B**), KEGG RIG-I LIKE RECEPTOR SIGNALING (**C**) and REACTOME DDX58 IFIH1 MEDIATED INDUCTION OF INTERFERON ALPHA BETA (**D**) gene signatures. (**E**,**F**) BT-549/tet-MUC1shRNA cells were treated with vehicle or DOX for 7 days. MUC1-C, RIG-I and MDA5 mRNA levels were analyzed by qRT-PCR (**E**). The results (mean ± SD of 4 determinations) are expressed as relative mRNA levels compared to that obtained for vehicle-treated cells (assigned a value of 1). The asterisk (*) denotes a *p*-value < 0.05. Lysates were immunoblotted with antibodies against the indicated proteins (**F**, **left**). Lysates from BT-549/CshRNA and BT-549/MUC1shRNA#2 were immunoblotted with antibodies against the indicated proteins (**F**, **right**). (**G**,**H**). BT-549/tet-MUC1shRNA cells treated with vehicle or DOX for 7 days and stimulated with poly I:C for 12 h (**G**) or IFN-β for 4 h (**H**) were analyzed for RIG-I and MDA5 mRNA levels by qRT-PCR. The results (mean ± SD of 4 determinations) are expressed as relative mRNA levels compared to that obtained for vehicle-treated cells (assigned a value of 1). Uncropped Western Blots can be found at Appendix A.

**Figure 3 cancers-14-02580-f003:**
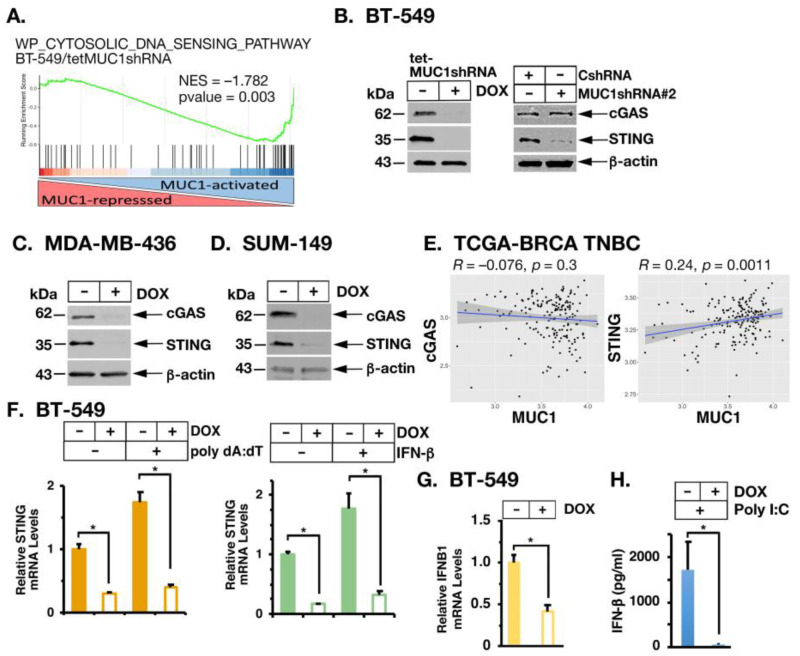
MUC1-C induces cGAS and STING expression. (**A**) Candidate enrichment plot of BT-549 RNA-seq data for the WP CYTOSOLIC DNA SENSING PATHWAY gene signature. (**B**) Lysates from (i) BT-549/tet-MUC1shRNA cells treated with vehicle or DOX for 7 days (**left**) and (ii) BT-549/CshRNA and BT-549/MUC1shRNA#2 cells (**right**) were immunoblotted with antibodies against the indicated proteins. (**C**,**D**) Lysates from MDA-MB-436/tet-MUC1shRNA (**C**) and SUM149/tet-MUC1shRNA (**D**) cells treated with vehicle or DOX for 7 days were immunoblotted with antibodies against the indicated proteins. (**E**) Scatter plots showing correlations of MUC1 with cGAS and STING in TNBCs from the TCGA-BRCA cohort. (**F**) BT-549/tet-MUC1shRNA cells treated with vehicle or DOX for 7 days and transfected with poly dA:dT for 12 h (**left**) or stimulated with IFN-β for 24 h (**right**) were analyzed for STING mRNA levels by qRT-PCR. (**G**) BT-549/tet-MUC1shRNA cells treated with vehicle or DOX for 7 days were analyzed for IFNB1 mRNA levels by qRT-PCR. The results (meanSD of 4 determinations) are expressed as relative mRNA levels compared to that obtained for vehicle-treated cells (assigned a value of 1). (**H**) BT-549/tet-MUC1shRNA cells were treated with vehicle or DOX for 7 days and then transfected with poly I:C for 12 h. Supernatants were analyzed for IFN-β levels by ELISA. The results (mean ± SD of 3 determinations) are expressed as IFN-β pg/mL. The asterisk (*) denotes a *p*-value < 0.05. Uncropped Western Blots can be found at Appendix A.

**Figure 4 cancers-14-02580-f004:**
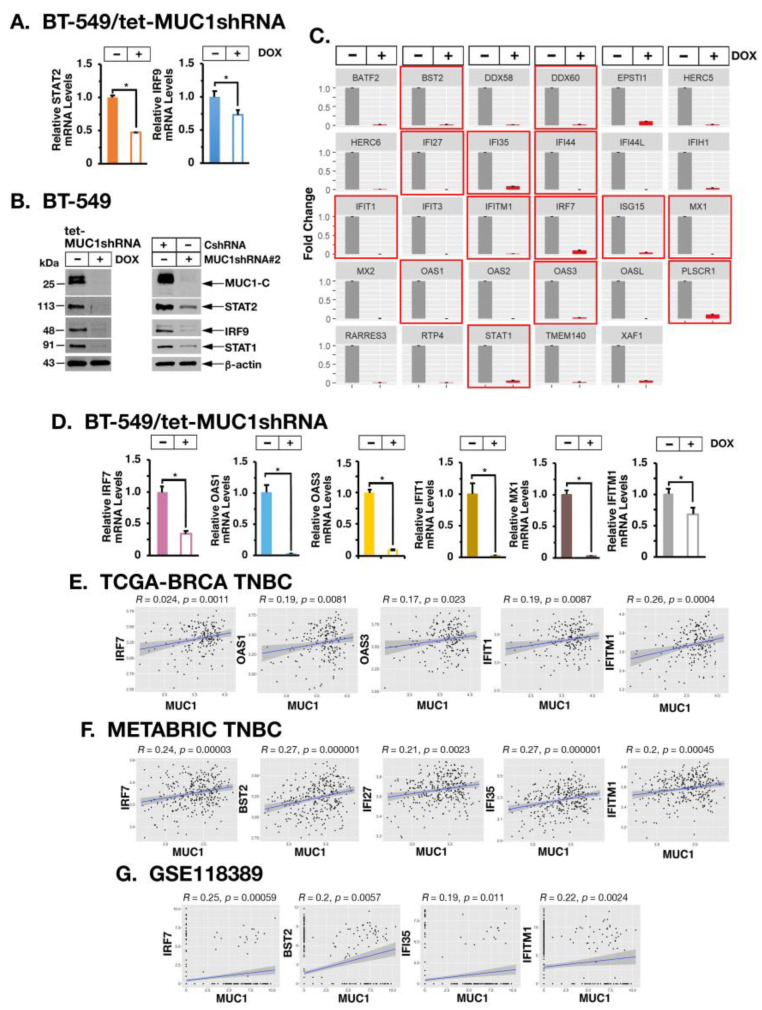
MUC1-C induces expression of U-ISGF3 target genes. (**A**) BT-549/tet-MUC1shRNA cells treated with vehicle or DOX for 7 days were analyzed for STAT2 and IRF9 mRNA levels by qRT-PCR. The results (mean ± SD of 4 determinations) are expressed as relative mRNA levels compared to that obtained for vehicle-treated cells (assigned a value of 1). (**B**) Lysates from (i) BT-549/tet-MUC1shRNA cells treated with vehicle of DOX for 7 days and (ii) BT-549/CshRNA and BT-549/MUC1shRNA#2 cells (**right**) were immunoblotted with antibodies against the indicated proteins. (**C**)**.** RNA-seq performed in triplicate on BT-549/tet-MUC1shRNA treated with vehicle (gray bars) or DOX (red bars) for 7 days was analyzed for expression of the indicated U-ISGF3 target genes. The results are expressed as the mean ± SD of 3 determinations. IRDS genes are highlighted with red boxes. (**D**) BT-549/tet-MUC1shRNA cells treated with vehicle or DOX for 7 days were analyzed for the indicated IRDS mRNA levels by qRT-PCR. The results (mean ± SD of 3 determinations) are expressed as relative mRNA levels compared to that obtained for vehicle-treated cells (assigned a value of 1). (**E**,**F**) Scatter plots showing correlations of MUC1 with the indicated IRDS genes in TNBCs from the TCGA-BRCA (**E**) and METABRIC (**F**) cohorts. (**G**) Scatter plots of the correlation between MUC1 and the indicated IRDS genes in single TNBC cells as analyzed from the GSE118389 scRNA-seq dataset. The asterisk (*) denotes a *p*-value < 0.05. Uncropped Western Blots can be found at Appendix A.

**Figure 5 cancers-14-02580-f005:**
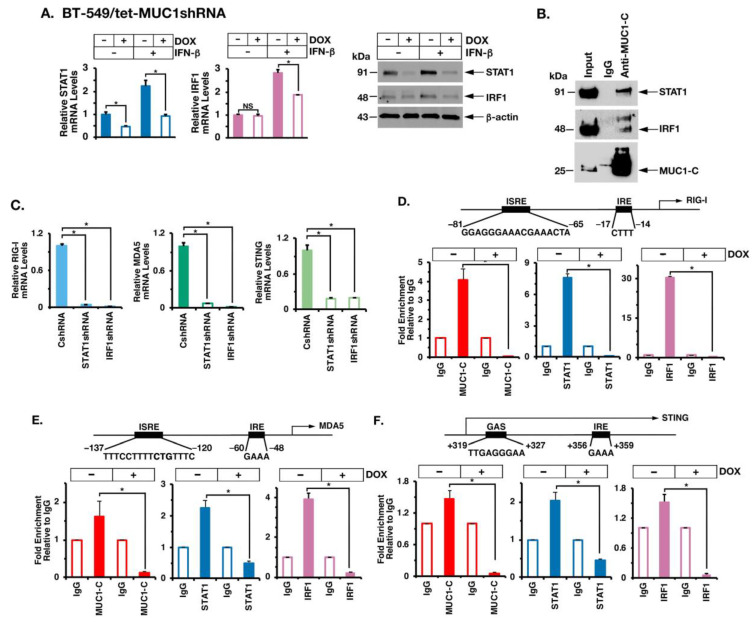
MUC1-C forms a nuclear complex with STAT1 and IRF1 in activating RIG-I, MDA5 and STING. (**A**) BT-549/tet-MUC1shRNA cells treated with vehicle or DOX for 7 days and stimulated with IFN-β for 4 h were analyzed for STAT1 and IRF1 mRNA levels by qRT-PCR (left). The results (mean ± SD of 4 determinations) are expressed as relative mRNA levels compared to that obtained for vehicle-treated cells (assigned a value of 1). Lysates were immunoblotted with antibodies against the indicated proteins (right). (**B**) Nuclear lysates from BT-549 cells stimulated with IFN-β for 4 h were precipitated with anti-MUC1-C and a control IgG. Input lysate and the precipitates were immunoblotted with antibodies against the indicated proteins. (**C**) BT-549 cells expressing a control CshRNA, STAT1shRNA or IRF1shRNA were analyzed for the indicated mRNA levels by qRT-PCR. The results (mean ± SD of 4 determinations) are expressed as relative mRNA levels compared to that obtained for the CshRNA cells (assigned a value of 1). (**D**–**F**) Schemas of the RIG-I (**D**), MDA5 (**E**) and STING (**F**) promoter regions with highlighting localization of ISREs, IREs and GAS. Soluble chromatin from BT-549/tet-MUC1shRNA cells treated with vehicle or DOX for 7 days was precipitated with a control IgG, anti-MUC1-C, anti-STAT1 and anti-IRF1. The DNA samples were amplified by qPCR with primers for the respective promoter regions. The results (mean ± SD of 3 determinations) are expressed as fold enrichment relative to that obtained with the IgG control (assigned a value of 1). The asterisk (*) denotes a *p*-value < 0.05. Uncropped Western Blots can be found at Appendix A.

**Figure 6 cancers-14-02580-f006:**
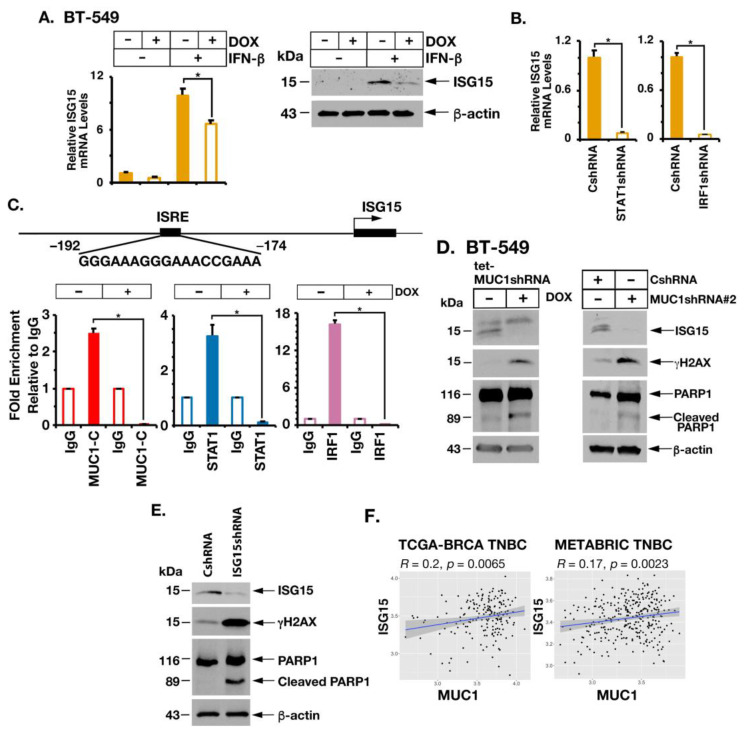
Silencing MUC1-C downregulates ISG15 expression and induces the DNA damage response. (**A**) BT-549/tet-MUC1shRNA cells treated with vehicle or DOX for 7 days and stimulated with IFN-β for 24 h were analyzed for ISG15 mRNA levels by qRT-PCR (**left**). The results (mean ± SD of 4 determinations) are expressed as relative mRNA levels compared to that obtained for vehicle-treated cells (assigned a value of 1). Lysates were immunoblotted with antibodies against the indicated proteins (**right**). (**B**) BT-549 cells expressing a CshRNA, STAT1shRNA or IRF1shRNA were analyzed for ISG15 mRNA levels by qRT-PCR. The results (mean ± SD of 4 determinations) are expressed as relative mRNA levels compared to that obtained for vehicle-treated cells (assigned a value of 1). (**C**) Schema of the *ISG15* promoter region with highlighting of the ISRE. Soluble chromatin from BT-549/tet-MUC1shRNA cells treated with vehicle or DOX for 7 days was precipitated with a control IgG, anti-MUC1-C, anti-STAT1 and anti-IRF1. The DNA samples were amplified by qPCR with primers for the *ISG15* promoter region. The results (mean ± SD of 3 determinations) are expressed as fold enrichment relative to that obtained with the IgG control (assigned a value of 1). (**D**) Lysates from (i) BT-549/tet-MUC1shRNA cells treated with vehicle or DOX for 7 days (**left**) and (ii) BT-549/CshRNA and BT-549/MUC1shRNA#2 (**right**) were immunoblotted with antibodies against the indicated proteins. (**E**) Lysates from BT-549 cells expressing a CshRNA or ISG15shRNA were immunoblotted with antibodies against the indicated proteins. (**F**) Scatter plots showing correlations of MUC1 with ISG15 in TNBCs from the TCGA-BRCA (**left**) and METABRIC (**right**) cohorts. The asterisk (*) denotes a *p*-value < 0.05. Uncropped Western Blots can be found at Appendix A.

**Figure 7 cancers-14-02580-f007:**
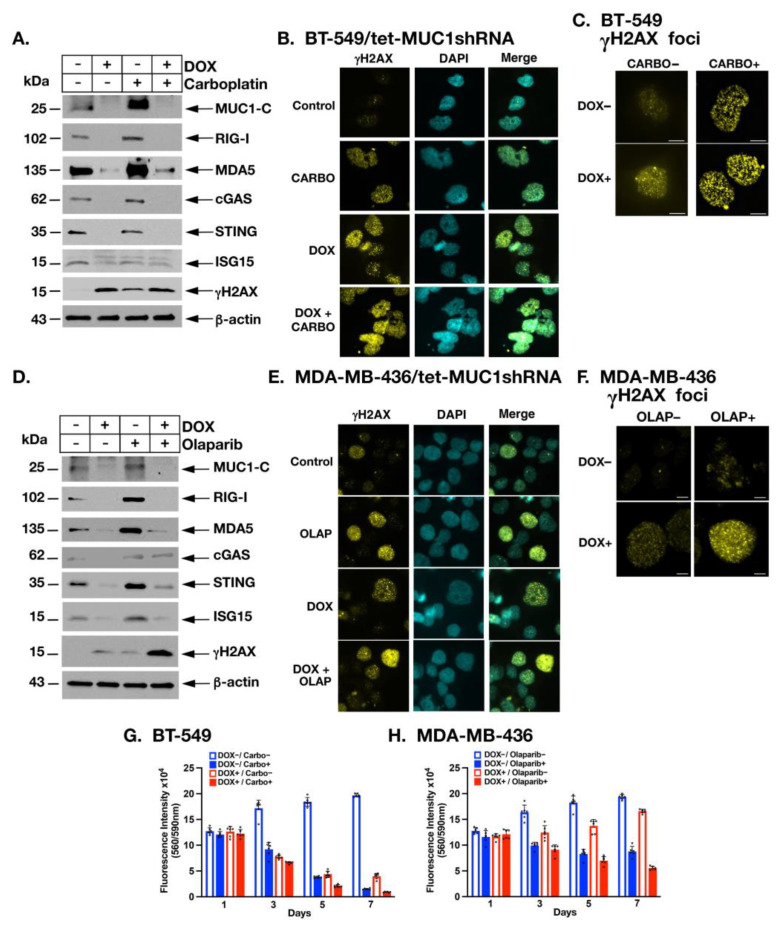
MUC1-C is necessary for carboplatin- and olaparib-induced activation of the cGAS/STING axis. (**A**) BT-549/tet-MUC1shRNA cells were treated with vehicle or DOX for 7 days and with or without 20 μM carboplatin for 48 h. Lysates were immunoblotted with antibodies against the indicated proteins. (**B**,**C**) BT-549/tet-MUC1shRNA cells were treated with vehicle or DOX for 7 days and with or without 20 μM carboplatin for 48 h. ICC analysis of γH2AX in the indicated cells (**B**) and at higher magnification for visualization of γH2AX foci (**C**; bar 2 μm). (**D**) MDA-MB-436/tet-MUC1shRNA cells were treated with vehicle or DOX for 7 days and with DMSO as the vehicle control or 5 μM olaparib for 72 h. Lysates were immunoblotted with antibodies against the indicated proteins. (**E**,**F**) ICC analysis of γH2AX in the indicated cells (**E**) and at higher magnification for visualization of γH2AX foci (**F** bar 2 μm). (**G**,**H**) Cell viability of the indicated BT-549 (**G**) and MDA-MB-436 (**H**) cells treated for the indicated times was determined by the AlamarBlue assay. Fluorescence intensity is expressed as the mean ± SD of 6 determinataions. Uncropped Western Blots can be found at Appendix A.

## Data Availability

The data presented in this study are available in this article and the Appendix A.

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
