# Peer review of "Targeting MUC1-C Suppresses Chronic Activation of Cytosolic Nucleotide Receptors and STING in Triple-Negative Breast Cancer"

_cancers, 2022, doi:10.3390/cancers14112580_

Round 1

Reviewer 1 Report

This is an extremely comprehensive study demonstrating that MUC1-C can suppress chronic activation of nucleotide receptors as well as STING in cell lines of triple negative breast cancer. The concept of differential gene activation during an acute protective response versus in a setting of chronic inflammation is often overlooked in cancer biology studies making this study one of high interest. The mechanistic studies involving the cascade of genes with MUC1-C silencing are extremely comprehensive. Most interesting to this reviewer is how these pathways intersect with the use of carboplatin or the PARP inhibitor olaparib.

Minor points:

  1. It is suggested that data from supplemental figure S8 be incorporated into the manuscript since it demonstrates a biologic consequence of the mechanistic studies being described.
  2. The authors should decide how they would employ MUC1-C targeting agents in combination with either carboplatin or a PARP inhibitor, or both, in a specific clinical setting of TNBC.

Author Response

Manuscript ID: cancers-1687207
Reviewer #1

1. It is suggested that data from supplemental figure S8 be incorporated into
the manuscript since it demonstrates a biologic consequence of the
mechanistic studies being described.

RESPONSE: As suggested, we have incorporated the panels from previous Supplemental Fig. S8 into Fig. 7A-H.

2.  The authors should decide how they would employ MUC1-C targeting agents in combination with either carboplatin or a PARP inhibitor, or both, in a specific clinical setting of TNBC.

RESPONSE: As requested, we have included a description of how the anti-MUC1-C agents could be used in combination for the treatment of patients with TNBC (Discussion; lines 553-556).

Reviewer 2 Report

The authors present a study investigating the role of MUCI-C in activating the IFN-gamma pathway and thereby, its involvement in DNA damage response. The study is well thought- the hypothesis is scientifically strong, and the experiments follow a logical sequence. However, a major concern/drawback is the presentation. Generally, for such studies a minimum of two different cell lines and 2 different shRNAs are needed to confirm their findings. The authors have conducted some experiments in multiple cell lines and some only in in BT-549. It would be helpful to rearrange their figures, such that either  data from 2 cell lines are together in the main manuscript, or both the shRNAs together. This would help the readers know that the findings are robust. In the present form, it is difficult to delineate which findings have been confirmed in multiple cell lines?

Additionally, please label your shRNAs. It is unlcear whether 2 shRNAs were used against STAT1 and IRF1?

Have the authors studied MUC1 translocation post the treatments as variable?

The authors have described one of their cell lines being BRCA mutant. Is there a rationale for it, because the authors don't say anything about SUM149 cell line. If this is to rule out the involvement of BRCA mutations, then that could be written into the text.

Minor comment: Line 239: Please introduce MDA5.

Author Response

Manuscript ID: cancers-1687207

Reviewer #2

  1. It would be helpful to rearrange their figures, such that either data from 2 cell lines are together in the main manuscript, or both the shRNAs together. This would help the readers know that the findings are robust.

  As requested, we have rearranged the figures by including results obtained from using both MUC1shRNAs in the same panels, specifically as shown in new (i) Fig. 2F, left and right, (ii) Fig. 3B, left and right, (iii) Fig. 4B, left and right, and (iv) Fig. 6D, left and right. 

  1. Additionally, please label your shRNAs. It is unclear whether 2 shRNAs were used against STAT1 and IRF1?

  In response, we have clearly labeled second shRNAs as shRNA#2. Two shRNAs were used in the experiments demonstrating the novel effects of silencing MUC1 and ISG15. STAT1 and IRF1 are well established upstream effectors of the type I IFN pathway. Accordingly, we only used single STAT1 and IRF1 shRNAs to confirm previous work. 

  1. Have the authors studied MUC1 translocation post the treatments as variable?

   Previous studies have demonstrated that MUC1-C translocates to the nucleus in the response to DNA damage. A statement to this effect has been included in the Discussion (line 362-363).

  1. The authors have described one of their cell lines being BRCA mutant. Is there a rationale for it, because the authors don't say anything about SUM149 cell line. If this is to rule out the involvement of BRCA mutations, then that could be written into the text.

   As suggested, we have stated that studies with BRCA mutant TNBC cells demonstrate that the reported MUC1-C regulated signaling pathways are independent of BRCA1 status (lines 146-149 and 189-191).

Minor comment: Line 239: Please introduce MDA5.

   As requested, we have introduced MDA5 in the text (lines 274-275).

Round 2

Reviewer 2 Report

Hello,

The revised version reads much better and the figures are more clearly presented. However, in Figure 7, the authors have used only a single shRNA and a single cell line, BT-549 to show the carboplatin and olaparib induced activation of the cGAS/STING axis. The authors need to validate these findings in a second shRNA. Ideally, data is presented in multiple cell lines and in 2 shRNAs. Since the data here is from a single cell line, a second shRNA is required to rule out any non-specific target effects.

Best wishes
